# Herpes Simplex Virus 1 and 2 Infections during Differentiation of Human Cortical Neurons

**DOI:** 10.3390/v13102072

**Published:** 2021-10-14

**Authors:** Petra Bergström, Edward Trybala, Charlotta E. Eriksson, Maria Johansson, Tugce Munise Satir, Sibylle Widéhn, Stefanie Fruhwürth, Wojciech Michno, Faisal Hayat Nazir, Jörg Hanrieder, Soren Riis Paludan, Lotta Agholme, Henrik Zetterberg, Tomas Bergström

**Affiliations:** 1Department of Psychiatry and Neurochemistry, Institute of Neuroscience and Physiology, Sahlgrenska Academy, University of Gothenburg, SE-405 30 Gothenburg, Sweden; petra.bergstrom@gu.se (P.B.); tugce.munise.satir@gu.se (T.M.S.); stefanie.fruhwurth@gu.se (S.F.); faisal.nazir@neuro.uu.se (F.H.N.); lotta.agholme@gu.se (L.A.); 2Department of Infectious Diseases, Institute of Biomedicine, Sahlgrenska Academy, University of Gothenburg, SE-413 46 Gothenburg, Sweden; edward.trybala@microbio.gu.se (E.T.); charlotta.c.eriksson@gmail.com (C.E.E.); maria.johansson@microbio.gu.se (M.J.); sibylle.widehn@telia.com (S.W.); 3Department of Rheumatology and Inflammation Research, Institute of Medicine, Sahlgrenska Academy, University of Gothenburg, SE-413 46 Gothenburg, Sweden; srp@biomed.au.dk; 4Department of Psychiatry and Neurochemistry, Institute of Neuroscience and Physiology, Sahlgrenska Academy, University of Gothenburg, SE-431 80 Mölndal, Sweden; wojciech.michno@neuro.gu.se (W.M.); jorg.hanrieder@neuro.gu.se (J.H.); henrik.zetterberg@clinchem.gu.se (H.Z.); 5Department of Neuroscience, Physiology and Pharmacology, University College London, London WC1E 6BT, UK; 6Department of Neurodegenerative Disease, Institute of Neurology, University College London, Queen Square, London WC1N 3BG, UK; 7Department of Biomedicine, Aarhus University, 8000 Aarhus, Denmark; 8UK Dementia Research Institute at University College London, London WC1E 6BT, UK; 9Clinical Neurochemistry Laboratory, Sahlgrenska University Hospital, SE-431 80 Mölndal, Sweden

**Keywords:** herpes simplex virus, human induced pluripotent stem cells, human neuroprogenitors, central nervous system infection, cortical neurons

## Abstract

Herpes simplex virus 1 (HSV-1) and 2 (HSV-2) can infect the central nervous system (CNS) with dire consequences; in children and adults, HSV-1 may cause focal encephalitis, while HSV-2 causes meningitis. In neonates, both viruses can cause severe, disseminated CNS infections with high mortality rates. Here, we differentiated human induced pluripotent stem cells (iPSCs) towards cortical neurons for infection with clinical CNS strains of HSV-1 or HSV-2. Progenies from both viruses were produced at equal quantities in iPSCs, neuroprogenitors and cortical neurons. HSV-1 and HSV-2 decreased viability of neuroprogenitors by 36.0% and 57.6% (*p* < 0.0001), respectively, 48 h post-infection, while cortical neurons were resilient to infection by both viruses. However, in these functional neurons, both HSV-1 and HSV-2 decreased gene expression of two markers of synaptic activity, CAMK2B and ARC, and affected synaptic activity negatively in multielectrode array experiments. However, unaltered secretion levels of the neurodegeneration markers tau and NfL suggested intact axonal integrity. Viral replication of both viruses was found after six days, coinciding with 6-fold and 22-fold increase in gene expression of cellular RNA polymerase II by HSV-1 and HSV-2, respectively. Our results suggest a resilience of human cortical neurons relative to the replication of HSV-1 and HSV-2.

## 1. Introduction

Herpes simplex virus 1 (HSV-1) and 2 (HSV-2) are double-stranded DNA viruses of the human alpha-herpesvirus sub-family, consisting of HSV-1, HSV-2, and varicella zoster virus (VZV). All three viruses establish latent infections in sensory neural ganglionic cells from where they may later reactivate to cause lytic infections in innervated keratinocytes. On rare occasions, these viruses induce lytic CNS infections with severe damage to cortical neurons as result, but the responsible pathogenetic mechanisms are not well understood. Regarding natural infection of HSV, several autopsy studies show that HSV-1 DNA can be found in the brains of individuals without any symptoms of infection [1,2]. In contrast, little data are available for the presence of HSV-2 in the CNS of subjects without symptoms and signs of HSV brain infection [3], but this virus may on rare occasions cause persistent brain infections also in immunocompetent individuals [4,5]. 

In adults, the most common manifestation of lytic HSV-1 CNS infection is herpes simplex encephalitis (HSE), a severe disease mainly located in the limbic system and the temporal lobe [6], while HSV-2 infection of the CNS typically results in a milder infection of the meninges, herpes simplex meningitis (HSM) [7,8]. Adult HSE patients have a 10–15% mortality rate and high incidence of severe neurological sequelae despite available antiviral treatment [9,10,11,12]. Although HSV-2 can cause myelitis and rare cases of encephalitis in adults, HSM patients most commonly return to normal health within two weeks after the onset of symptoms with no or minor neurological deficits [7,13]. In contrast, in neonates, HSV-2 can result in a severe CNS infection with pathogenic effects that exceed those of HSV-1 both in regards to inflammatory response, symptoms and prognosis [14,15]. Why HSV-1 and HSV-2 CNS infections differ in these manners is still unknown, but viral as well as host factors may contribute to the differences [16,17]. 

Lack of relevant cell models to study HSV-1 and HSV-2 infections in the CNS has hampered the understanding of their neuropathology. Recently, it has been demonstrated that human induced pluripotent stem cells (iPSCs) and iPSC-derived CNS neurons can be infected by HSV-1 and used in latency models [18,19,20], while HSV infection during normal cortical neuronal differentiation has not been described in detail and HSV-2 infection in iPSCs or developing cortical neurons has not yet been studied. 

Here, we present a cell model for studies of HSV infection during the differentiation of human iPSCs into neuroprogenitor cells (NPCs) and further to functional, cortical neurons. In this cell model, primary and secondary neuronal progenitor cells are successively generated, followed by the formation of deep-layer cortical neurons. After 90 days, neurons from both deep and upper layers of the cortex are present in the cultures and have formed functional synaptic networks [21,22,23,24,25]. Two low-passaged neurovirulent clinical strains were used to infect the cells: HSV-1 2762 isolated from the brain of a patient with fatal HSE during a clinical trial of acyclovir treatment [11,16] and HSV-2 VF-1181 obtained from the cerebrospinal fluid (CSF) of a patient with HSM [8,16]. 

By using this model, we investigated if cortical neurons are vulnerable to HSV-1 and HSV-2 infections during development, with a focus on cytopathogenicity and effects on axonal integrity and synaptic activity and plasticity. We found that even though viral replication and infectivity of both HSV-1 and HSV-2 remained stable during differentiation, infection with both viruses showed most cytopathic effects in neuronal progenitors, while infection of cortical neurons had low effects on cell viability at the same time point after infection. We also found that HSV-2 was significantly more cytopathic than HSV-1 relative to both iPSCs and NPCs, but this difference in cytopathogenicity was less obvious in functional, cortical neurons. In neurons, both viruses decreased synaptic activity, as well as expression of genes involved in synaptic activity or synaptic organization and plasticity, without affecting axonal integrity.

## 2. Materials and Methods

### 2.1. Differentiation of iPSCs towards Cortical Neurons

Differentiation of cortical neurons was performed on three fibroblast-derived human iPSC lines (Ctrl1 [26], ChiPSC22 (Takara Bio Europe, Gothenburg, Sweden) and WTSIi015-A (EBiSC, Salisbury, UK), as described previously [21,22,24]. Briefly, iPSCs were cultured in a humidified atmosphere at 5% CO_2_ and 37 °C in complete Essential 8 (E8) medium on Geltrex coat (both from Thermo Fisher Scientific, Waltham, MA, USA) or in Complete mTeSR1 medium (Stem Cell Technologies, Saint-Egrève, France) on Matrigel coat (Corning, Corning, NY, USA). One day prior to neuronal induction, the iPSCs were passaged using EDTA (Thermo Fisher Scientific, Waltham, MA, USA) and pooled 2:1 to reach high density. The following day, the differentiation towards cortical neurons was initiated by changing medium to neural maintenance medium (NMM) consisting of DMEM/F12 and neurobasal medium (1:1) supplemented with 1 × N2 supplement, 1 × B27 supplement, 50 µM 2-mercaptoethanol, 0.5 × non-essential amino acids, 100 µM L-glutamine (all from Thermo Fisher Scientific, Waltham, MA, USA), 2500 U/mL penicillin/streptomycin (GE Healthcare, Chicago, IL, USA), 10 µg/mL insulin and 0.5 mM sodium pyruvate (both from Sigma-Aldrich, Saint Louis, MO, USA). NMM was further supplemented with 500 ng/mL mouse Noggin/CF chimera (R&D Systems, Minneapolis, MN, USA ) and 10 µM SB431542 (Stemgent, Cambridge, MA, USA). The cells were maintained in NMM for 10–12 days until a neuroepithelial cell layer had formed. The neuroepithelial cells were then dissociated in colonies using 10 mg/mL Dispase II (Thermo Fisher Scientific, Waltham, MA, USA) and replated on laminin-coated plates (1–2 µg/cm^2^; Sigma-Aldrich) in NMM supplemented with 20 ng/mL FGF2 (PeproTech Nordic, Stockholm, Sweden). The cells were kept in FGF2-supplemented medium for 4 to 5 days and then further passaged with dispase two times before day 25 in order to prevent differentiation towards other cell types. After 25 days, the colonies were dissociated into single cells using StemPro Accutase (Thermo Fisher Scientific, Waltham, MA, USA) and then further passaged with accutase once or twice before day 35. On day 35, the cells were passaged for the final time and re-plated onto plates coated with 0.01% poly-l-ornithine (Sigma-Aldrich, Saint Louis, MO, USA) and 1–2 µg/cm^2^ laminin in NMM. The cells were then maintained for up to two months, with medium changes every second day. Neuronal maturation was assessed with immunocytochemistry and qPCR using cortex-specific markers during differentiation, as described previously [21,22]. 

### 2.2. Viruses

Two low-passaged neurovirulent clinical strains were used to infect the cells in all experiments. HSV-1 2762 was isolated from the brain of a patient with fatal HSE during a clinical trial of acyclovir treatment [11], and HSV-2 VF-1181 was isolated from the CSF of a patient with HSM [8]. Both viruses were previously defined in terms of neurovirulence in comparison to several other HSV strains [16].

### 2.3. Preparation of Cells for Viral Infection

All cells were grown on Corning^®^ Primaria™ 24-well cell culture plates (VWR International, Spånga, Sweden), µ-Slide 8 well chamber slides (Ibidi, Gräfelfing, Germany) or multielectrode array plates (Multichannel Systems, Reutlingen, Germany). Three time points were chosen as illustrative of iPSC, NPC and neuron stages. For infection of iPSCs, the cells were passaged using EDTA (Thermo Fisher Scientific, Waltham, MA, USA) and replated onto Geltrex (Thermo Fisher Scientific, Waltham, MA, USA)-coated plates. For infection of NPCs, cells differentiated for 30 days were passaged using accutase and replated onto laminin-coated plates. For infection of neurons, NPCs were passaged on day 35 of differentiation using accutase and replated onto poly-l-ornithine/laminin-coated plates at a density of 50,000 cells/cm^2^ and differentiated to cortical neurons in NMM for another 55 days.

### 2.4. Infection of Cells during Neuronal Differentiation

The cells were infected with HSV-1 2762 or HSV-2 VF-1181 at three differentiation stages: iPSCs (day 0), NPCs (day 30) and cortical neurons (day 90) at a multiplicity of infection (MOI) of 0.1 PFU/cell and 1 h adsorption time. After 1 h incubation with virus at 37 °C in a humidified atmosphere with 5% CO_2_, the cells were carefully washed once, and fresh medium was added to all wells. Samples were collected immediately after medium change (0 h), or after 24 or 48 h of incubation. For long-term experiments, the cells were washed once directly following infection, and full medium changes were then carefully performed every 48 h without washing. Samples for viral replication were collected every 24 h, as well as immediately following each medium change. Samples for cellular viability and gene expression were collected six days post-infection. Non-infected control cells were prepared simultaneously and treated in a similar fashion, and at least two different iPSC lines were used in each experiment. 

### 2.5. Immunocytochemistry

Cells grown on Ibidi µ-slides (Ibidi, Gräfelfing, Germany) were fixed with Histofix for 20 min at room temperature (Histolab Products, Västra Frölunda, Sweden), covered with PBS and stored in fridge at 4 °C until analysis. The cells were permeabilized using 0.1% Triton-X100 in DPBS and blocked in block buffer (0.1 M glycine, 2% BSA, 0.1% Triton-X100 in PBS). Primary antibodies were diluted in block buffer (anti-OCT4 1:200 (Cell Signaling Technology, Danvers, MA, USA D73C4), anti-NANOG 1:400 (Cell Signaling Technology, Danvers, MA, USA C30A3), anti-TBR1 1:300 (Abcam, Cambridge, UK ab31940), anti-CTIP2 1:300 (Abcam, Cambridge, UK ab18465), anti-SATB2 1:100 (Abcam, Cambridge, UK ab51502), anti-TUJ1 1:1000 (Abcam, Cambridge, UK ab14545) and anti-S100 1:400 (Dako, Santa Clara, CA, USA Z0311) and incubated at 4 °C overnight. The cells were washed and blocked again and then incubated with Alexa488-conjugated and Alexa647-conjugated secondary antibodies (Thermo Fisher Scientific, Waltham, MA, USA) for 2 h at room temperature. From this point, the samples were protected from light, and the cells were stained with 1 µM DAPI (Thermo Fisher Scientific, Waltham, MA, USA). After repeated washes with PBS, the cells were mounted using Ibidi mounting medium (Ibidi, Gräfelfing, Germany). The samples were analyzed using a Nikon EclipseTi (Nikon Instruments, Amstelveen, Netherlands) inverted fluorescent microscope with 10× objective, and images were captured using the DU-897 Andor camera and the Nis Elements software (Nikon, Amstelveen, Netherlands). Alternatively, samples were analyzed using a Zeiss LSM700 inverted confocal microscope with 40–63× objectives and the ZEN2000 software (Carl Zeiss, Oberkochen, Germany). Image analysis was performed using ImageJ (Research Services Branch at NIH).

### 2.6. Extraction of Total RNA

At time points 0, 24 and 48 h after infection, the supernatant was removed from infected and uninfected cells from the three differentiation stages. The cells were washed once with PBS (room temperature) and then lysed in RLT Plus buffer (AllPrep DNA/RNA Mini kit/RNeasy Mini or Micro kits, Qiagen, Hilden, Germany) supplemented with dithiothreitol (4 mM; Sigma-Aldrich, Saint Louis, MO, USA). Lysates were immediately stored at −80 °C and thawed on shaker before manual extraction of total RNA or on a QIAcube robotic work station (Qiagen, Hilden, Germany) using AllPrep DNA/RNA Mini kit/RNeasy Mini or Micro columns, (Qiagen, Hilden, Germany) according to instructions provided by the manufacturer. Concentrations of total RNA samples were measured on a NanoDrop 2000/2000 c (Thermo Fisher Scientific, Waltham, MA, USA) in triplicates. 

### 2.7. Quantitative PCR

cDNA was synthesized from 250 to 500 ng of total RNA using a High Capacity cDNA kit with RNase inhibitor (Thermo Fisher Scientific, Waltham, MA, USA) in a total reaction volume of 20 µL and converted in a single-cycle reaction on a 2720 Thermal Cycler (Thermo Fisher Scientific) at 25 °C for 10 min, 37 °C for 120 min and 85 °C for 5 min. Quantitative PCR was performed using inventoried TaqMan Gene Expression Assays with FAM reporter dye in TaqMan Universal PCR Master Mix with UNG according to manufacturer’s instructions but in a total reaction volume of 25 µL. qPCR reactions were performed on Micro-Amp 96-well optical microtiter plates on a 7900 HT Fast QPCR System (Thermo Fisher Scientific, Waltham, MA, USA) using standard settings for Standard Curve qPCR. TaqMan Gene Expression Assays (all from Thermo Fisher Scientific, Waltham, MA, USA) for the following genes were used: POU class 5 homeobox 1 (*POU5F1*/*OCT4*: Hs01895061_u1); paired box 6 (*PAX6*: Hs00242217_m1); eomesodermin (*EOMES/TBR2*: Hs00172872_m1); T-box, brain, 1 (*TBR1*: Hs00232429_m1); B-cell CLL/lymphoma 11B (*BCL11B/CTIP2*: Hs01102259_m1); calcium/calmodulin-dependent protein kinase II beta (*CAMK2B*: Hs00365799_m1); allograft inflammatory factor 1 (*AIF1*: Hs00610419_g1); myelin basic protein (*MBP*: Hs00921945_m1); activity regulated cytoskeleton associated protein (*ARC*: Hs01045540_g1); ribosomal protein L27 (*RPL27*: Hs03044961_g1); ribosomal protein L30 (*RPL30*: Hs00265497_m1); hypoxanthine phosphoribosyltransferase 1 (*HPRT1*: Hs02800695_m1); eukaryotic 18S rRNA (*18S*: Hs03003631_g1); and RNA polymerase II subunit A (POLR2A: Hs00172187_m1). The amount of 2.5 ng cDNA was used in the PCR, and all samples were run in duplicates. First Strand cDNA Human Brain (Nordic Biosite, Täby, Sweden) was used as positive control. PCR results were analysed with the SDS 2.3 software (Thermo Fisher Scientific, Waltham, MA, USA), and the relative quantity of gene expression was determined using the ∆∆CT method [27], with *RPL27* or *18S* as endogenous reference for differentiations and with *18S* for infected versus non-infected. Both *RPL27* and *18S* gene expressions were stable during differentiation. The expression of *18S* was stable after infection with both HSV-1 and HSV-2. As a calibrator in the calculations, the average ΔCT values of iPSCs (*OCT4*), neurons (*PAX6*, *TBR2*, *TBR1* and *CTIP2*) or control samples (*CAMK2B* and *ARC; POLR2A*) were used. To adjust for variations between differentiations, all RQ values were divided by the average RQ of the calibrators and set to 1. 

### 2.8. Electron Microscopy

iPSCs, NPCs, and cortical neurons growing on plastic inserts in 24 well plates were infected with HSV-1 or HSV-2 at a MOI of 3, and the infected cells were rinsed twice and incubated in 0.5 mL of fresh NMM in 5% CO_2_-atmosphere at 37 °C for 42 h or until the development of viral cpe, i.e., for 16 h (iPSCs), for 24 h (NPCs) or for 24–42 h (neurons). The cells were fixed by adding 0.5 mL of 2.5% glutaraldehyde and incubated for 30 min in the CO_2_ incubator. The fixative was then collected, and the cells were fixed again with 2.5% glutaraldehyde in 0.02 M Soerensen buffer for 1 h at room temperature. The cells were rinsed triple with 50 mM Tris-HCl buffer, pH 7.4, and then processed for electron microscopy as described by Widehn and Kindblom [28]. Electron microscopy observations were carried out on a Philips CM10 transmission electron microscope, where images were captured using the iTEM Image Analysis Platform (Olympus, Tokyo, Japan), or a Focused Ion-Beam combined with a scanning electron microscope, FIB-SEM workstation, through annular bright-field scanning transmission electron microscopy, BF-STEM analysis (GAIA3 Tescan). The instrument was operated in the UH resolution scan mode at 30.0 kV, with WD of 5.4 mm, yielding a spot size of approximately 1.6 nm and with an effective pixel size of approximately 2.5 nm at 100,000× magnification.

### 2.9. Cell Viability Assay

Cells at three differentiation stages (iPSCs, NPCs or neurons) were infected with HSV-1 or HSV-2 or mock-infected in triplicate as described above. GMK AH-1 cells and human fibroblasts were infected at the same MOI (0.1 PFU/cell) for control experiments. After 48 h incubation in a CO_2_ atmosphere at 37 °C, CellTiter 96^®^ AQueous One Solution reagent (Promega Biotech, Madison WI, USA) containing an MTS salt (3-(4,5-dimethyl-2-yl)-5-(3-carboxymethoxyphenyl)-2-(4-sulfophenyl)-2 H-tetrazolium) was added to the cells at a dilution factor of 1:5 in cell medium and incubated in a CO_2_-atmosphere at 37 °C for 2 h. The supernatant from each well was then transferred to triplicate wells on a non-irradiated, flat-bottomed 96-well Nunc plate (Thermo Fisher Scientific, Waltham, MA, USA ), and absorbance was measured in a Multiscan FC microplate reader (Thermo Fisher Scientific, Waltham, MA, USA) at 492 nm.

### 2.10. Sample Preparation for Virus Plaque Assay and DNA Quantification

After one hour incubation with virus, the cells and cell-conditioned medium were collected from separate wells, either directly after infection (0 h) or after 24 or 48 h incubation in fresh medium. The infected cells were subjected to a rapid freeze-thaw cycle in −80 °C ethanol and 37 °C water bath in order to release the infectious virus. The samples (medium and cells) were stored at −80 °C until analysis. Human fibroblasts were used as controls. Data are presented as the total amount of infectious virus and DNA in each well.

### 2.11. Viral Plaque Assay for Viral Titer Determination

For viral plaque assays, African green monkey kidney (GMK-AH1) cells [29] were infected with serial 10-fold dilutions of the virus samples prepared from HSV-infected iPSCs, NPCs and neurons. Infected cells were incubated for 1 h in 5% CO_2_ atmosphere at 37 °C. The inoculum was then removed, the cells were rinsed and covered with methylcellulose and incubated in 5% CO_2_ at 37 °C for 3 days, followed by staining and fixation of cells with crystal violet. The viral plaques were counted to calculate the titer of produced infectious viral particles expressed in plaque forming units (PFU)/mL. 

### 2.12. Viral DNA Purification and Quantification

Viral DNA was purified in a MagNA Pure LC robot (Roche Diagnostics, Basel, Switzerland) using the MagNA Pure LC DNA isolation kit 1 (Roche Diagnostics, Basel, Switzerland) according to the manufacturer’s instructions. The HSV DNA was quantified after PCR amplification as previously described [30]. Briefly, respective glycoprotein B (gB)-regions of HSV-1 and HSV-2 were amplified using a pair of type-specific primers and probes. This resulted in the detection and amplification of a 118-nucleotide segment of the respective gB-regions. Here, specific cycle threshold (C_t_) values were related to a virus-specific standard curve to assess the number of DNA copies/mL in each sample. Procedures were carried out using a 7300 Real-Time PCR System (Thermo Fisher Scientific, Waltham, MA, USA). 

### 2.13. Immunochemical Quantification of Tau and NfL

Cell medium was collected 48 h after infection or, to serve as a positive control for axonal damage after a single dose of 4.5 Gy ionizing radiation, centrifuged at 400× *g* for 5 min to remove cell debris and frozen in −80 °C until analysis. Medium concentrations of total tau were measured by using INNOTEST^®^ hTAU Ag ELISA (Fujirebio, Tokyo, Japan) according to protocol. Briefly, biotinylated HT7 and BT2 antibody solutions and samples were added to an AT120 mAb strip plate in duplicates. The plate was agitated for 1 min at 700 rpm and incubated overnight at RT. The following day, the wells were washed 5 × 30 s with 1 × wash buffer before incubation with a Peroxidase-labelled streptavidin. After a second wash, the wells were incubated with tetramethylbenzidine (TMB) substrate solution for 30 min at RT kept from light. Stop reagent (0.9 N H_2_SO_4_) was then added directly to the wells to stop the reaction. The plate was mixed and the absorbance was quantified at 450 nm using a Multiskan FC microplate reader (Thermo Fisher Scientific, Waltham, MA, USA) within 15 min. NfL concentrations were measured in the cell medium using NF-light^®^ (Neurofilament light) ELISA from UmanDiagnostics, Billerica, MA, USA, according to manufacturer’s instructions. In short, the Anti NF-light strip plate was washed with 1× wash buffer before addition of duplicate samples, diluted 1:1 in sample diluent and incubated for 1 h at RT with 800 rpm agitation. After washing, the wells were incubated with biotin anti-NF-L mAb for 45 min, washed again and incubated with streptavidin-HRP conjugate for 30 min. After a final wash, the wells were incubated with TMB for 15 min before stop reagent (8% *v/v* H_2_SO_4_) was added to the wells. The absorbance was quantified at 450 nm with 620 nm reference wavelength at a Multiskan FC microplate reader (Thermo Fisher Scientific, Waltham, MA, USA).

### 2.14. Multielectrode Array Analysis of Synaptic Activity

Cells were differentiated towards cortical neurons according to Shi et al. [24] for 35 days before plating on poly-L-ornithine-coated (0.01%; Sigma-Aldrich, Saint Louis, MO, USA) and laminin-coated 60–6 well MEAs plates (Multi Channel Systems, Reutlingen, Germany) in independent culture chambers with 9 electrodes in each chamber. Ten days prior to analysis, NMM was replaced by BrainPhys medium to enhance synaptic function [31]. The cells were infected with HSV-1 or HSV-2 at MOI = 1 for 1 h. Measurements were performed as previously described [22] immediately prior to infection and 6 and 24 h post-infection. The cells were monitored under the microscope immediately before recordings to assure that the electrodes were covered by cells. A number of spikes (action potentials) were used to determine changes in overall neuronal activity.

### 2.15. Statistical Methods

GraphPad Prism 6.05 (GraphPad Software, San Diego, CA, USA) was used for graph construction and statistical analysis. Student’s unpaired t-test was used to compare means when the samples approximated a normal distribution. Mann–Whitney U test was used when the samples did not approximate a normal distribution. *p*-values < 0.05 were considered significant. 

## 3. Results

### 3.1. Differentiation of iPSCs towards Cortical Neurons

Based on our previous observations [21,22], three differentiation time points were chosen to represent important cortical neuronal differentiation stages. These included human iPSCs before the neuronal differentiation was initiated, cultures differentiated for one month consisting of a mixture of primary and secondary progenitor cells (NPCs) and cultures differentiated for three months, consisting of a mixture of NPCs and neurons from both deep and upper cortical layers with functional synaptic networks [21,22,23,24,25]. Representative phase-contrast images of the successive differentiation from round iPSCs to pyramidal neurons with increasing neurite networks are shown in Figure 1A. Representative confocal images of cells stained with markers specific to different cortical developmental stages are shown in Figure 1B. The presence of the stem cell markers Nanog homeobox (NANOG) and POU class 5 homeobox 1 (POU5F1/OCT4) confirmed the iPSC identity before differentiation (left panels). After 30–50 days of differentiation, paired box 6 (PAX6)-positive cortical rosettes appeared and an increasing number of B-cell CLL/lymphoma 11B (BCL11B /CTIP2)-positive layer V neurons formed from the edges of the cortical rosettes (middle panels). After 90 days of differentiation, staining of neuron-specific class III beta-tubulin (TUJ1) showed extensive axonal outgrowth, and a small proportion of S100 calcium-binding protein B (S100)-positive astrocytes appeared at this point. Both T-box, brain, 1 (TBR1)-positive deep-layer neurons and SATB homeobox 2 (SATB2)-positive upper-layer neurons were present in the neuronal cultures (right panels). The cell identity was confirmed with qPCR markers specific for iPSCs, NPCs and cortical neurons, respectively (Figure 1C). The stem cell marker OCT4 (i) was expressed in iPSCs but disappeared when the cells were differentiated to NPCs and neurons. PAX6 (ii), a marker for primary neuronal progenitors, appeared in NPCs and continued to increase during differentiation towards cortical neurons. Eomesodermin (EOMES/TBR2) (iii), a marker for secondary neuronal progenitors, could be detected in iPSCs but increased in NPCs and neurons. CTIP2 (iv), a marker for cortical layer V neurons, and TBR1 (v), a marker for cortical deep-layer neurons, appeared in NPC cultures and increased during differentiation. The cultures were qPCR-negative for the microglia marker allograft inflammatory factor 1 (AIF1) and the oligodendrocyte marker myelin basic protein (MBP) (vi). More detailed validations of the cell model are described in [21,22].

### 3.2. Morphological Features of HSV-1 and HSV-2 Infected Cells during Neuronal Differentiation 

Next, iPSCs, NPCs and neurons were infected with HSV-1 2762 or HSV-2 VF-1181 clinical strains and investigated by electron microscopy (EM). A striking feature of iPSCs (Figure 2A–C), independent of viral infection, was the presence of voluminous intracytoplasmic vacuoles (marked v), possibly formed by the fusion of individual lipid droplets. Infection of iPSCs with HSV-1 (Figure 2B) or HSV-2 (Figure 2C) induced cell rounding already at 16 h post infection. Uninfected NPCs (Figure 2D) exhibited long but relatively wide cytoplasmic extensions (asterisks), of which some contained parallel threads of microtubules (mt) (Figure 2D, inset). After twenty-four hours post-infection of NPCs with both HSV-1 (Figure 2E) and HSV-2 (Figure 2F), these cytoplasmic extensions were no longer evident in the infected cultures. Uninfected cortical neurons (Figure 2G) exhibited long and slender cytoplasmic extensions (Figure 2G, asterisks), with clearly detectable microtubules (Figure 2G inset, mt). Both HSV-1 (Figure 2H) and HSV-2 virions (Figure 2I) (white arrows) were frequently found attached to, rather than inside, these extensions. After 42 h post-infection with HSV-1 and 24 h post-infection with HSV-2, the neuronal extension remained relatively intact with clearly detectable microtubules (Figure 2H,I, mt).

In iPSCs, envelopment of HSV-1 virions (Figure 3A, white arrows) seemed to occur by penetration of virus capsids (white arrowheads) into the lumen of large vacuoles in the cytoplasm. The same phenomenon was not observed for envelopment of HSV-2 (Figure 3B), where mature virions (white arrows) were found in the cytoplasm but outside the vacuoles. In the cytoplasm of NPCs infected with HSV-1, virions were frequently detected in tubular structures (Figure 3C, black arrows in insets), while HSV-2 virions were detected in the cytoplasm (Figure 3D, white arrow) and not within the tubular structures observed for HSV-1. In cortical neurons, both HSV-1 (Figure 3E,G) and HSV-2 (Figure 3F,H) virions (white arrows) were found in the cytoplasm. Viral capsids in the cytoplasm or nucleus are marked with white arrow heads. Egress of both HSV-1 (Figure 3G) and HSV-2 (Figure 3H) virions (white arrows) in cortical neurons seemed to occur in tubulo-vesicular structures or pits close to the soma membrane.

### 3.3. Cytopathogenicity of HSV-1 and HSV-2 during Neuronal Differentiation

In order to further investigate if the cells were particularly sensitive to HSV infection at any specific stage of neuronal differentiation, as suggested by the EM images, we measured cell viability of iPSCs, NPCs and neurons 48 h after infection with HSV-1 2762 or HSV-2 VF-1181 clinical strains. HSV is known to cause selective shut-off of host protein synthesis in order to use cellular resources for the production of viral nucleic acids and proteins. Consequently, following the production of progeny virus particles, cellular metabolism and, thus, viability are severely compromised. To study this, we used a tetrazolium salt, which dehydrogenizes to a colored product in mitochondria of viable metabolically active cells. The color change was quantified as absorbance and related to non-infected control cells. 

While no significant reduction in iPSC viability was observed after infection with HSV-1, HSV-2 infection slightly reduced the cell viability of iPSCs compared with non-infected controls (14.6%) (Figure 4A). In NPCs, both HSV-1 and HSV-2 infections significantly reduced cell viability compared to non-infected controls (36.0% and 57.6% reduction, respectively) (Figure 4B), with effects comparable to or even exceeding those observed for human embryonic fibroblasts used here as non-neuronal control cells (Figure 4D). In both iPSCs and NPCs (Figure 4A and Figure 4B, respectively), the decrease in cell viability was significantly greater in cells infected with HSV-2 than in cells infected with HSV-1. In neurons, both HSV-1 and HSV-2 reduced cell viability but to a lesser extent than in NPCs (15.8% and 20.8%, respectively) (Figure 4C). No significant difference in cell viability was observed between HSV-1- and HSV-2-infected neurons. In summary, both viruses decreased cell viability most in NPCs compared with both iPSCs and neurons. HSV-2 decreased viability more profoundly than HSV-1 in iPSCs and NPCs, which is in line with previous reports using animal models or cell lines showing pronounced cytopathic effect after HSV-2 infections [16,32]. On the contrary, no significant differences in cell viability were observed between HSV-1 or HSV-2 infected functional neurons.

### 3.4. Viral Infectivity Remains Stable during Cortical Neuron Differentiation

To investigate whether the observed difference in HSV cytopathogenicity during neuronal differentiation was the result of different capabilities of the viruses to productively infect iPSCs, NPCs and cortical neurons, both viral replications (measured as viral DNA copies/mL) and the formation of infectious particles (measured as plaque-forming units [PFU]/mL) were quantified 0, 24 and 48 h post-inoculation. We found that, immediately after the medium change following inoculation, both HSV-1 and HSV-2 DNA quantities were significantly lower in NPCs than in neurons (Figure 5A,C, 0 h) and that HSV-1 and HSV-2 PFU were significantly lower in NPCs than in both iPSCs and neurons (Figure 5B,D, 0 h). 

Twenty-four hours post-infection, both HSV-1 DNA quantities (Figure 5A, 24 h) and replicative yield assayed as PFU (Figure 5B, 24 h) were significantly higher in neurons than in iPSCs or NPCs; however, after 48 h, no differences were observed between the three differentiation stages (Figure 5A,B, 48 h). 

The HSV-2 DNA quantity was significantly higher in iPSCs than in NPCs 24 h post-infection (Figure 5C, 24 h), while the HSV-2 yield in the form of PFU followed the pattern observed for HSV-1 with significantly higher numbers of PFU in neurons than in both iPSCs and NPCs (Figure 5D, 24 h). Forty-eight hours post-infection, the HSV-2 DNA quantity was significantly higher in iPSCs than in both NPCs and neurons (Figure 5C, 48 h), while, again, no differences were observed in the HSV-2 PFU yield between the three differentiation stages (Figure 5D, 48 h). We cannot explain this discrepancy, but it could be suggested that the production of mature HSV-2 virions from viral DNA is more efficient in NPCs and neurons than in iPSCs. 

Together, these data suggest that the production of HSV-1 and HSV-2 infectious virions is similar between iPSCs, NPCs and neurons 48 h post-infection and that the differences in cytopathogenicity observed between the cell stages are due to other factors rather than viral load.

### 3.5. HSV Infection Does Not Induce Markers of Axonal Damage

The finding that neurons coped well with HSV infection was somewhat surprising considering the general notion that cytopathogenicity of these viruses is pronounced in most non-neuronal cells. To investigate if HSV infection had other effects on vital neuronal features, such as axonal integrity, we measured secreted neurofilament light (NfL) and tau, two well-established markers of axonal damage (reviewed in [33,34]), in the cell-conditioned medium after infection. In our cell model, NfL (Figure 6A) and tau (Figure 6B) are normally present at clearly measurable levels in neuron-conditioned cell medium but not in media from iPSCs or NPCs. Interestingly, no difference in NfL (Figure 6C) or tau (Figure 6D) secretion was observed in HSV-infected cells compared to non-infected control neurons, while a single radiation insult increased secretion of both proteins compared to non-irradiated control (Figure 6E,F). The fact that these markers did not increase in the cell-conditioned medium indicated sustained integrity of axons in neurons infected with HSV.

### 3.6. HSV Infection Decreases Expression of Genes Related to Synaptic Activity and Plasticity

We have previously shown that the gene expression of calcium/calmodulin-dependent protein kinase II beta (CAMK2B) and activity regulated cytoskeleton associated protein (Arc) correlates with synaptic function in our neuronal model [22]. CAMK2B is involved in axonal organization and glutamatergic synaptic density [35,36]. ARC is a neuronal activation immediate-early gene. ARC mRNA is acutely increased upon synaptic activation [37,38], and ARC gene expression is, therefore, widely used as a marker of synaptic activation in animal models (reviewed in [39]). To investigate if HSV infection would affect CAMK2B and ARC expression in cortical neurons, mRNA of these two genes were measured 48 h after HSV infection. Here, viral infections with both HSV-1 and HSV-2 significantly decreased the gene expression of CAMK2B in cortical neurons (Figure 7A). Both HSV-1 and HSV-2 decreased ARC expression in neurons, although only the effect of HSV-1 on this mRNA reached statistical significance (Figure 7B), indicating that HSV infections affected genes involved in synaptic activity and plasticity. 

Next, we explored the effects of HSV infection on synaptic activity using multielectrode array (MEA). Neurons were differentiated in MEA plates with electrode-covered bottoms until they showed synaptic activity, as we have previously described [22]. The neurons were then infected with HSV-1 or HSV-2 for 1 h and synaptic activity was measured before HSV infection, as well as 6 and 24 h post-infection (Figure 7C,D). HSV-1 decreased synaptic activity already after 6 h. Twenty-four hours after infection with both HSV-1 (Figure 7C) and HSV-2 (Figure 7D), no synaptic activity could be detected. No recordings were performed 48 h post-infection since cells starting to detach from the electrodes at this point could have impact on the results.

### 3.7. HSV Replication Continues during Long-Term Infection of Cortical Neurons

In order to investigate how HSV infection would progress over time in functional neuronal cells, cortical neurons differentiated for 90 days were infected with HSV-1 or HSV-2 at MOI = 0.1 and followed for six days, with full medium changes every second day (Figure 8). HSV DNA copies in the medium were measured daily, as well as directly after medium changes (MC) on day 0, day 2 and day 4 (Post-MC d0/Post-MC d2/Post-MC d4). Both HSV-1 (Figure 8A) and HSV-2 (Figure 8B) DNA increased after each medium change, indicating sustained viral replication throughout the six day long post-infection period. 

When cell viability was measured in the cultures six days post-infection, the overall viability was 47% and 26% of non-infected controls for HSV-1 and HSV-2, respectively (Figure 8C), resembling the pattern observed in NPCs 48 h post-infection. The transcription of HSV-1 mRNA has been shown to require a functional and active cellular RNA polymerase [40,41,42]. Therefore, we measured mRNA levels of RNA polymerase II subunit A (POLR2A), encoding the largest subunit of the polymerase responsible for mRNA transcription in eukaryotic cells, in both HSV-1 and HSV-2 infected neurons at 6 days post-infection. We found that POLR2A expression increased 6-fold and 22-fold by HSV-1 and HSV-2, respectively, when compared with non-infected control neurons (Figure 8D). This finding indicates a transcriptional upregulation of the POLR2A gene also after long-term HSV infection of cortical neurons.

## 4. Discussion

The neuropathogenicity of HSV-1 and HSV-2 and the influence of neuronal differentiation on HSV virulence largely remain to be elucidated. Novel models of human iPSCs primed towards NPCs and differentiated onwards to cortical neurons provide opportunities to specifically study viral CNS infections during neuronal differentiation in vitro. The protocol used in this study spans three months and closely follows the cortical neuronal differentiation in the uterus regarding the timing of distinct stages of maturation [23,25]. We used this cell model to infect human cells with two clinical CNS-derived HSV strains with demonstrated neuropathogenicity: one HSV-1 strain derived from the brain of a patient with HSE [11,16] and one HSV-2 strain derived from CSF of a patient with HSM [8,16]. The cells were infected with these two viruses at three defined stages of neuronal differentiation—iPSCs, NPCs and cortical neurons—to investigate viral replication and production of infectious virions in the different cell types, as well as effects of the infection on cell viability and neuronal features.

One main finding was that, at a stage before an elaborate neurite network with active synapses has formed [21,25], cell viability in NPCs was markedly affected by HSV infection. At this stage of neuronal development, the decrease in cell viability by HSV was comparable to that of human fibroblasts. At later stages when neuronal cell morphology had developed, cell viability was less affected by infection, and virus-producing cells were still present six days post-infection. The finding indicates that human cortical neurons can handle HSV infection relatively well in a non-cytopathic manner, while immature neuronal cells are more vulnerable.

In iPSCs and NPCs, investigation with EM suggested differences between HSV-1 and HSV-2 regarding the cytoplasmic sites of viral envelopment, while in cortical neurons envelopment and egress of both viruses seemed to occur in the cell soma and at the plasma membrane. Cell viability of both iPSCs and NPCs was also more profoundly affected by HSV-2 than by HSV-1, while no such differences were observed in cortical neurons 48 h post-infection. Neuronal morphology, a post-mitotic state, or general neuronal activities such as synaptic activity could possibly influence the vulnerability to HSV infections. 

The cell model used here, consisting predominantly of neuronal cells [23], not only provides a possibility to study the effect of HSV on cortical neurons without interference of other cell types but also to investigate the effect of HSV on neurons co-cultured for example with iPSC-derived microglia. The lower levels of cell death in neurons observed here is in line with the proposal that the cytopathic effects of HSV infection on neurons in HSE are mediated by infection of and/or inflammation induced by other cell types, such as oligodendrocytes, astrocytes or microglia, and not caused by the virus infection of cortical neurons per se. If confirmed, this may have relevance for the choice of therapeutic strategy to treat or prevent HSE.

The next question we posed was whether the observed cell stage-dependent cytopathogenicity was caused by differences in viral yields in developing and functional neurons. Our data revealed that the production of mature viral particles 48 h post-infection remained stable throughout neuronal differentiation; thus, there was no obvious link between productive infection of HSV and cell death in differentiated human neuronal cells. Instead, developmental maturation seemed to provide the neurons with efficient functions for virus replication and egress, with relatively low induction of cytopathogenicity over time and viable, virus-producing cells still present six days post-infection. In line with a recent study showing that cellular RNA polymerase is rapidly recruited to the HSV genome after infection of human cells to enable transcription of HSV genes [43], we observed increased transcription of cellular RNA polymerase in the cortical neurons after long-term infection by both viruses and more pronounced by HSV-2. The relation between increased transcription of this gene and productive HSV infection in cortical neurons is interesting, but more investigation is needed to determine whether this is a direct viral effect or a cellular response to the infection. 

In neurons, neither virus induced substantial cell death 48 h post-infection, while the decrease in viability in the neuronal cultures six days post-infection was comparable to the decrease observed in NPCs 48 h post-infection. However, six days after infection, the cultures still contained enough viable cells for sustained viral replication to take place. It is important to consider that although functional, the neurons used here represent young neurons during fetal development [25,44] and that the cultures contain a mixture of neurons and NPCs also after 90 days of differentiation. One possible explanation for the decreased viability six days post-infection is that the NPCs in the mixed cultures die from the infection, while either substantial numbers of infected neurons or few infected neurons producing large amounts of progeny virus were still viable and continued to produce virus at this point. This leaves open the possibility that functional neurons are more resistant to HSV-induced cell death, while developing neurons may be vulnerable to both HSV-1 and HSV-2 infections. 

Investigation using EM revealed signs of substantial morphological changes in HSV-infected iPSCs and NPCs, while the morphology of functional neurons, including microtubule-containing projections, seemed to remain relatively intact during the time window studied after infection. In our cell model, the axonal proteins NfL and tau are naturally and increasingly secreted in the cultures as the neuronal networks mature [22]. However, upon axonal damage both of these markers increase in CSF [33,34], as well as here in our neuron-conditioned medium, making them suitable as markers of neuronal damage in vitro. The observation that neither tau nor NfL secretion increased after infection is in line with the lack of morphological changes observed with EM and suggests that HSV infection had little effect on neuronal and axonal integrity. 

To assess if HSV would still affect neuronal functions, we first measured mRNA expression of CAMK2B and ARC, which are commonly used as markers of synaptic plasticity and activation [38]. We have previously linked mRNA expression of both CAMK2B and ARC to synaptic function in our cell model [22] and, interestingly, here both HSV-1 and HSV-2 decreased CAMK2B and ARC mRNA expression in cortical neurons 48 h post-infection. This suggests that synaptic activity and/or synaptic plasticity was negatively affected by the infection. However, the regulation of immediate-early genes and the roles of their gene products in synaptic activity and plasticity are complex and not yet fully understood [39]. For example, a recent study by Acuña-Hinrichsen and colleagues show increased Arc expression in mouse cortical neurons after HSV infection [45]. Furthermore, by using a mouse model of latent HSV infection, Doll et al. [46] found that reactivation of the virus resulted in neuronal destruction related to the viral DNA replication. Differences in experimental setups, such as MOI used or timing after infection, as well as differences between rodent and human neuronal responses could contribute to this discrepancy. However, in order to further investigate the effects of HSV infection on synaptic function, we measured synaptic activity using MEA technology. Supporting the fact that the decreased CAMK2B and ARC expressions reflected decreased synaptic function, both HSV-1 and HSV-2 infections decreased synaptic activity in the MEA experiments. In agreement with a recently published study showing decreased synaptic activity of human iPSC-derived neurons after HSV infection [47], these results suggest that even at a time point when HSV infection had low effects on viability and axonal integrity, neuronal function was impaired.

Taken together, we demonstrated here that cortical neurons were more resilient to HSV infections than cells at earlier stages of differentiation despite stable virus yields. Changes in morphology and function, such as the emergence and establishment of a neurite network, at later stages of neuronal differentiation may contribute to this observation. Future studies using this model will further address the impact of HSV-1/HSV-2 infections on neuronal functions, including synaptic activities. 

## Figures and Tables

**Figure 1 viruses-13-02072-f001:**
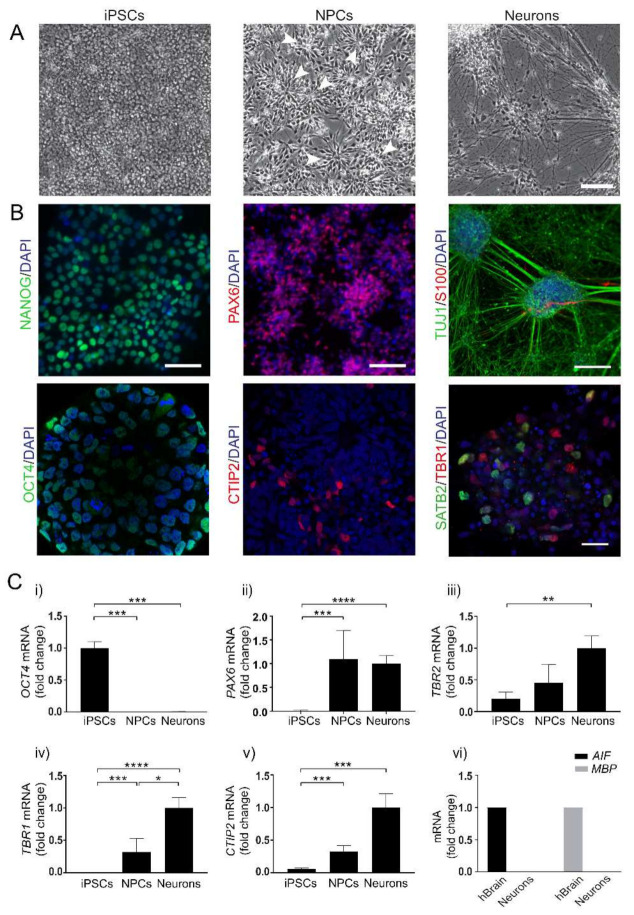
Differentiation of human iPSCs towards cortical neurons. (**A**) Representative phase contrast images showing cell morphology of human induced pluripotent stem cells (iPSCs) before differentiation, neuroprogenitor cells (NPCs) forming cortical rosette structures (arrows) after 30 days of differentiation and cortical neurons (Neurons) forming neurite networks after 90 days of differentiation. Scale bar = 100 µm. (**B**) Representative confocal images from immunocytochemistry staining of cells during neuronal differentiation; iPSCs stained positive for the stem cell markers NANOG (green, left upper panel) and OCT4 (green, left lower panel). In the NPC cultures, the cortical rosettes stain positive for primary progenitor marker PAX6 (red, upper mid-panel), and CTIP2-positive layer-V neurons (red, lower mid-panel) start to appear around the edges of the cortical rosettes. After three months of differentiation, immunocytochemistry staining of neuron-specific tubulin (TUJ1, green) shows extensive neurite networks in neurons, with a small fraction of S100-positive astrocytes (red) appearing (right upper panel). At this point, the cultures contain both TBR1-positive layer-VI neurons (red) and SATB2-positive upper-layer neurons (green) (right lower panel). Scale bar = 100 µm for TUJ/S100B, 50 µm for NANOG and PAX6 and 25 µm for remaining images. DAPI-stained nuclei are shown in blue. (**C**) qPCR analysis of differentiation markers shows expression of the pluripotency marker OCT4 in iPSCs but not in NPCs or Neurons (i). Gene expressions of PAX6 (primary neuron progenitors) (ii), TBR2 (secondary neuronal progenitors) (iii), TBR1 (layer VI neurons) (iv) and CTIP2 (layer V neurons) (v) are all present in NPCs and increase with differentiation. Bars represent mean of four separate differentiations for iPSCs and three separate differentiations for NPCs and neurons from two iPSC lines +/− SEM. * *p* < 0.05, ** *p* < 0.01,*** *p* < 0.001 and **** *p* < 0.0001 with Mann–Whitney U test. (vi) Markers for microglia (AIF1) and oligodendrocytes (MBP) were detected with qPCR in cDNA from human brain but not in cDNA preparations from cultured neurons (*n* = 2).

**Figure 2 viruses-13-02072-f002:**
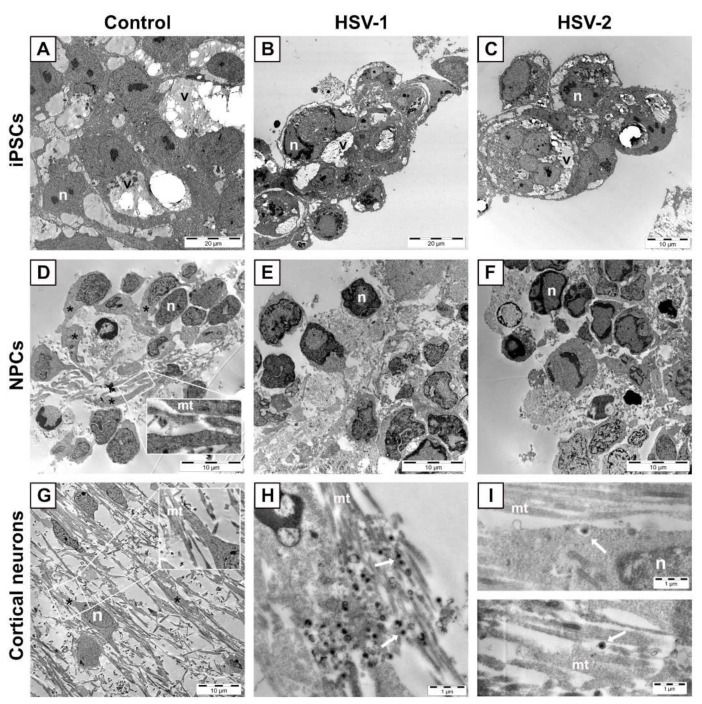
(**A**–**C**) Representative images of iPSCs uninfected or infected with HSV at MOI = 3. (**A**) Uninfected iPSCs. Large lipid-containing vacuoles (v) are observed in the cytoplasm of iPSCs, independent of infection. Infection of iPSCs with HSV-1 (**B**) or HSV-2 (**C**) induces cell rounding 16 h post-infection. (**D**–**F**) Representative images of NPCs uninfected or infected with HSV at MOI = 3. (**D**) Uninfected NPCs show long and wide extensions of the cytoplasm (asterisks) with clearly detectable microtubules (mt in inset) that are lost 24 h post-infection with HSV-1 (**E**) or HSV-2 (**F**). (**G**–**I**) Representative images of cortical neurons uninfected or infected with HSV at MOI = 3. Neurons display long projections (asterisk) containing microtubules (mt in, inset). These structures are relatively intact 42 h post-infection with HSV-1 (**H**) and 24 h post-infection with HSV-2 (**I**). *n*, nucleus; v, large lipid-containing vacuole.

**Figure 3 viruses-13-02072-f003:**
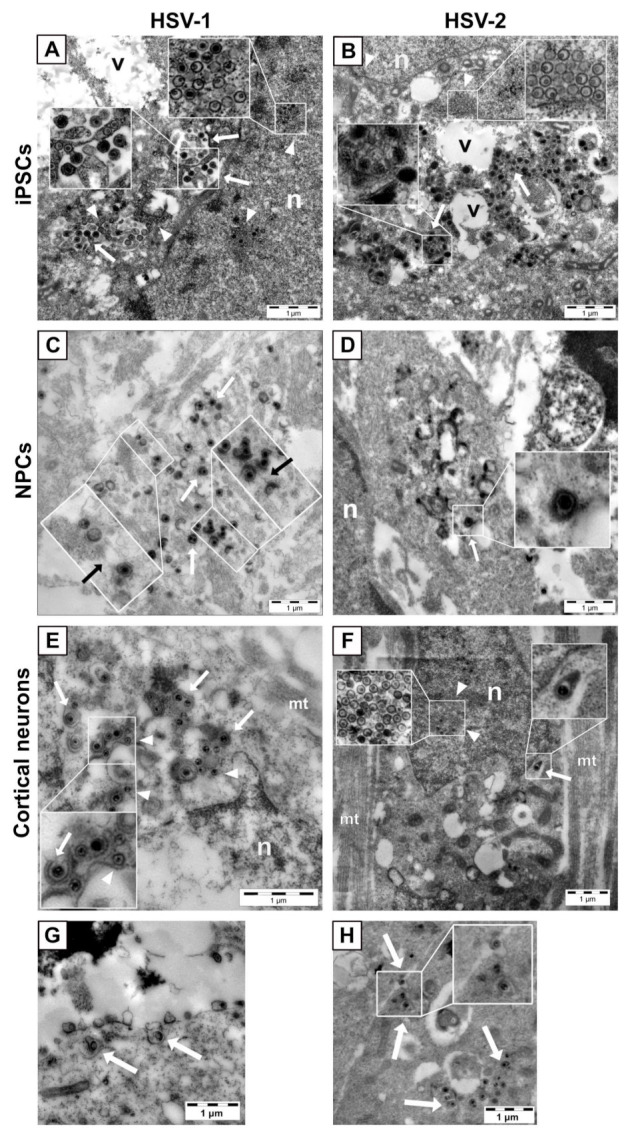
(**A**,**B**) Envelopment and egress of HSV in iPSCs, NPCs and neurons. Representative images of iPSCs infected with HSV-1 (**A**) or HSV-2 (**B**) at MOI = 3. In iPSCs, capsid envelopment of HSV-1 virions ((**A**), white arrows) occurs in large vacuoles in the cytoplasm, while HSV-2 virions ((**B**), white arrows) are observed in the cytoplasm outside the vacuoles. Viral capsids in the nucleus and in the cytoplasm are pointed out by arrowheads. (**C**,**D**) Representative images of NPCs infected with HSV-1 (**C**) or HSV-2 (**D**) at MOI = 3. In NPCs, envelopment of HSV-1 virions ((**C**), white arrows) is observed in tubular structures (**C**, black arrows in insets) in the cytoplasm. HSV-2 envelopment ((**D**), white arrow) is observed in the cytoplasm but not within the tubular structures observed for HSV-1. (**E**,**F**) Representative images of cortical neurons infected with HSV-1 (**E**) or HSV-2 (**F**) at MOI = 3. Viral capsids are marked with white arrowheads. Egress of both HSV-1 (**G**) and HSV-2 (**H**) particles is observed in tubulo-vesicular structures near the cell plasma membrane of the neuronal soma (white arrows). n, nucleus; v, large, lipid-containing vacuole; mt, microtubules.

**Figure 4 viruses-13-02072-f004:**
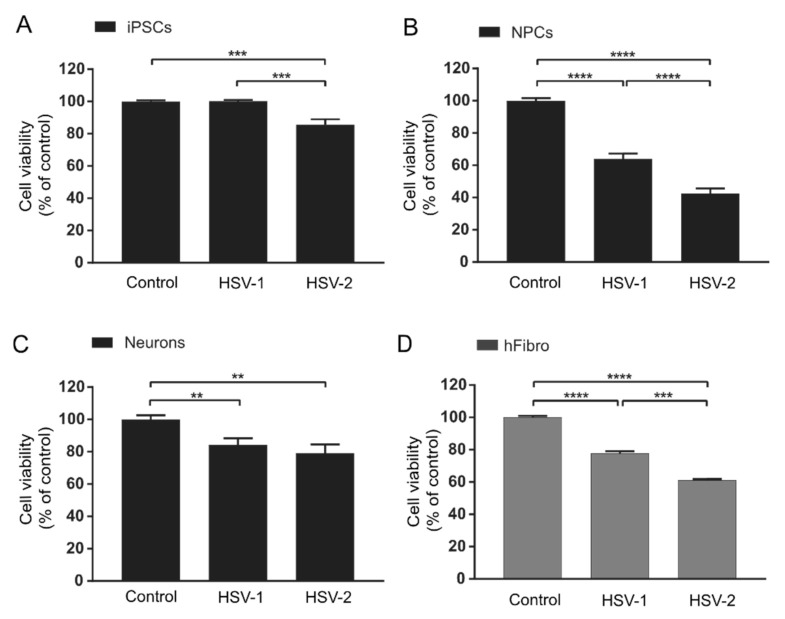
Viability of cells infected with herpes simplex virus (HSV) 1 and 2 at three stages during neuronal differentiation measured 48 h post-infection at MOI = 0.1. (**A**) HSV-1 infection does not affect the cell viability of iPSCs, while HSV-2 infection reduces cell viability of iPSCs by 14.6% compared with non-infected control iPSCs. Bars represent mean of six separate experiments on two iPSC lines +/− SEM. (**B**) HSV-1 and HSV-2 infections reduce cell viability of NPCs by 36.0% and 57.6%, respectively, compared with non-infected control NPCs. Bars represent mean of five separate differentiations from two iPSC lines +/− SEM. (**C**) Neuron viability is slightly reduced 48 h after infection (15.8% and 20.8% by HSV-1 and HSV-2, respectively). No significant difference in cytopathic effect is observed between HSV-1 and HSV-2 in neurons. Bars represent mean of four separate differentiations on two iPSC lines +/− SEM. Viability of human fibroblasts (**D**) infected with herpes simplex virus (HSV) 1 and 2 measured 48 h post-infection at MOI = 0.1. Bars represent mean of three separate experiments +/− SEM. ** *p* < 0.01, *** *p* < 0.001 and **** *p* < 0.0001 from Student’s *t*-test.

**Figure 5 viruses-13-02072-f005:**
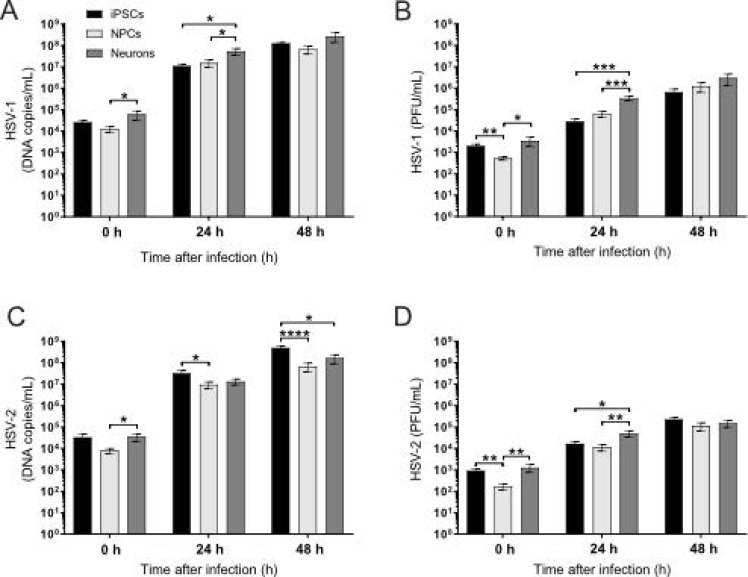
Viral DNA and PFU in iPSCs, NPCs and neurons, viral DNA copies/mL measured with qPCR and the amount of infectious virus measured as plaque-forming units (PFU/mL) 0, 24 or 48 hours post-inoculation with HSV-1 or HSV-2 at MOI = 0.1 (**A**–**D**). When measured directly after the medium change following infection, HSV-1 DNA copies/mL ((**A**), 0 h) is significantly lower in NPCs than in neurons, and HSV-1 PFU/mL ((**B**), 0 h) is significantly lower in NPCs than in both neurons and iPSCs. Twenty-four hours post-inoculation, HSV-1 DNA copies/mL ((**A**), 24 h) and HSV-1 PFU/mL ((**B**), 24 h) were significantly higher in neurons than in both iPSCs and NPCs. Forty-eight hours post-inoculation, no differences in HSV-1 DNA copies/mL ((**A**), 48 h) or HSV-1 PFU/mL ((**B**), 48 h) were observed between the three differentiation stages. At 0 h post-inoculation, HSV-2 DNA copies/mL (**C**) was significantly lower in NPCs than in neurons, and HSV-2 PFU/mL (**D**) was significantly lower in NPCs than in both neurons and iPSCs. Twenty-four hours post-inoculation, HSV-2 DNA copies/mL (**C**) was significantly higher in iPSCs than in NPCs, while HSV-2 PFU/mL was significantly higher in neurons than in both iPSCs and NPCs. Forty-eight hours post-inoculation, HSV-2 DNA (**C**, 48 h) was significantly higher in iPSCs than in NPCs and neurons, while no differences in HSV-2 PFU/mL (**D**) were observed between the three differentiation stages. iPSCs and NPCs; bars represent mean of six separate differentiations from two iPSC lines +/− SEM. Neurons; bars represent mean of four separate experiments from two iPSC lines +/− SEM. * *p* < 0.05, ** *p* < 0.01, *** *p* < 0.001 and **** *p* < 0.0001 from Student’s *t*-test.

**Figure 6 viruses-13-02072-f006:**
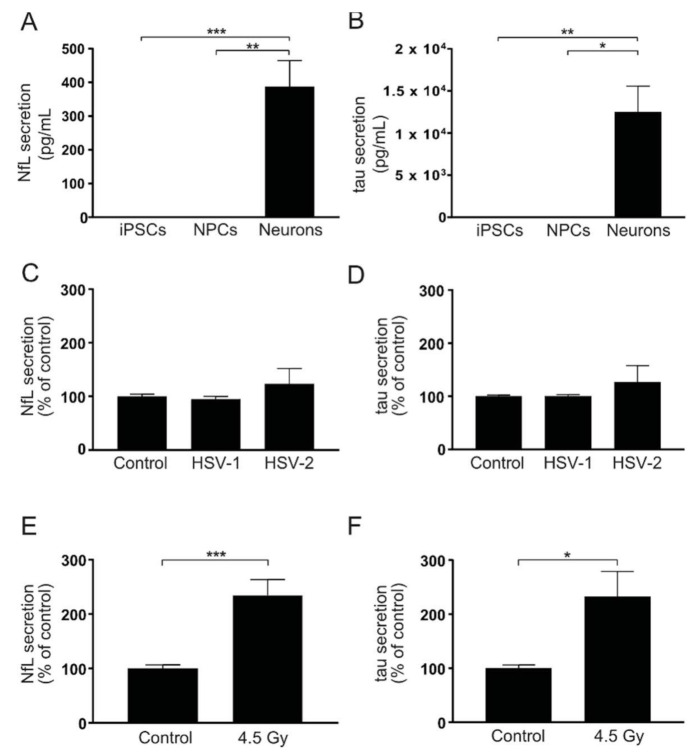
HSV infection does not induce secretion of axonal-injury markers in iPSC-derived cortical neurons. (**A**,**B**) NfL and tau measured in the cell-conditioned medium from iPSCs, NPCs and neurons. NfL (A) and tau (**B**) are actively secreted by neurons but not by iPSCs or NPCs. Bars represent mean of six (iPSCs), four (NPCs) or five (neurons) separate experiments from two iPSC lines +/− SEM. (**C**,**D**) NfL and tau measured in cell-conditioned medium from neurons 48 h after infection with HSV-1 or HSV-2 at MOI = 0.1. No significant differences in secretion of NfL (**C**) or tau (**D**) were observed between infected and non-infected neurons. Bars represent mean of three separate differentiations from two iPSC lines +/− SEM. (**E**,**F**) NfL and tau measured in the cell-conditioned medium from neurons after a single irradiation insult with 4.5 Gy. Both NfL (**E**) and tau (**F**) secretions increased significantly compared to control 48 h after irradiation. Bars represent mean of four separate differentiations from three iPSC lines +/− SEM. * *p* < 0.05, ** *p* < 0.01 and *** *p* < 0.001 from Student’s *t*-test.

**Figure 7 viruses-13-02072-f007:**
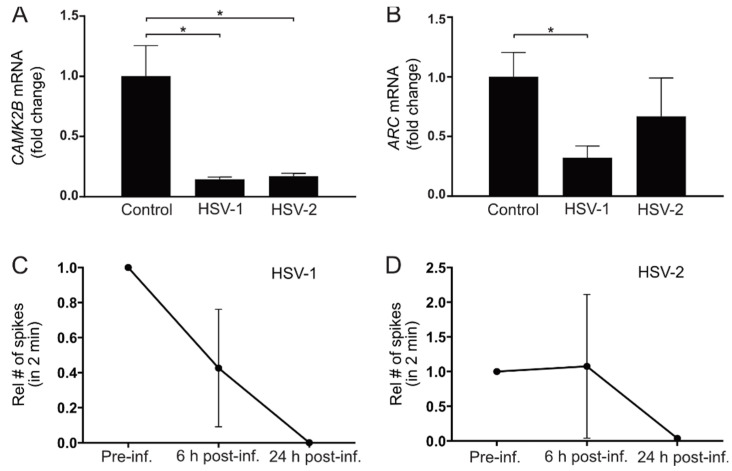
HSV infection of neurons reduces synaptic activity and gene expression related to synaptic activity and plasticity. (**A**,**B**) CAMK2B and ARC gene expression measured with qPCR in neurons 48 h after infection with HSV-1 or HSV-2 at MOI = 0.1. (**A**) Viral infection of neurons with HSV-1 and HSV-2 significantly decreased the expression of the synaptic plasticity marker CAMK2B. (**B**) Viral infection of neurons with HSV-1 significantly decrease expression of ARC compared with non-infected control neurons. HSV-2 infection shows a tendency to decrease ARC gene expression compared with control, although not reaching statistical significance. Target gene expression was related to reference gene 18S, which remains stable after infection, and correlated to average control. Bars represent mean fold change of three separate differentiations from two iPSC lines +/− SEM. * *p* < 0.05 from Student’s *t*-test. (**C**,**D**) Synaptic activity measured with multielectrode array in neurons 6 and 24 h after infection with HSV-1 or HSV-2 at MOI = 1. (**C**) HSV-1 decreased synaptic activity in neurons 6 and 24 h post-infection. (**D**) HSV-2 decreased synaptic activity in neurons twenty-four hours post-infection. Data represent the mean number of spikes in 2 min for two separate differentiations from one iPSC line +/− range relative to the respective well pre-infection (set to 1).

**Figure 8 viruses-13-02072-f008:**
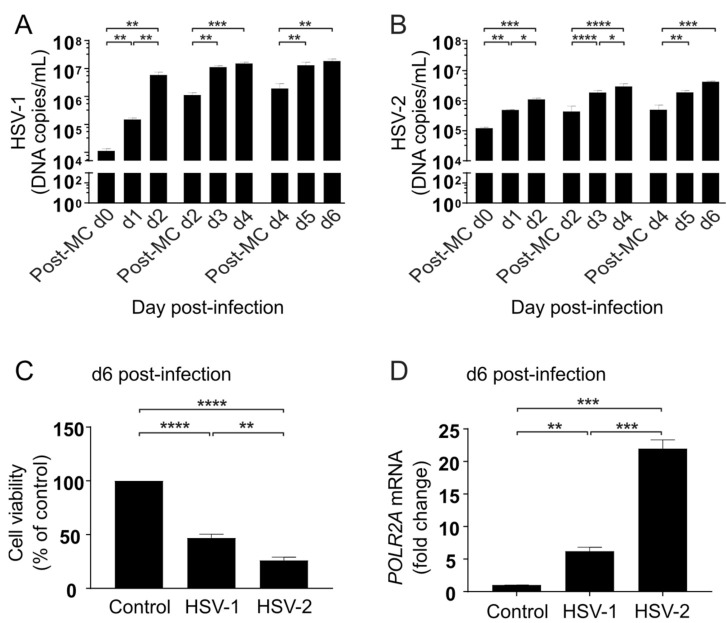
Neuronal responses followed six days after HSV infection. Ninety days old neurons infected for one hour with HSV-1 or HSV-2 at MOI = 0.1 and followed for six days post-infection. (**A**,**B**) HSV-1 DNA copies/mL measured with qPCR in the cell medium, directly after the medium change (MC) following inoculation (Post-MC d0) 24 h post-infection (d1) and 48 h post-infection (d2), shows that replication of HSV-1 (**A**) and HSV-2 (**B**) increases during the first two days (d0–d2). Viral DNA measured directly after the medium change 2 days post-infection (Post-MC d2) and again after 24 h (d3) and 48 h (d4) shows that replication continues 3–4 days post-infection. Viral DNA measured directly after the medium change 4 days post-infection (Post-MC d4) and again after 24 h (d5) and 48 h (d6) shows that replication continues 5–6 days post-infection. Bars represent mean of three separate differentiations from two iPSC lines +/− SEM. (**C**) Six days post-infection, neuron viability is reduced with 47.0% and 68.5% by HSV-1 and HSV-2, respectively. Bars represent mean of three separate differentiations from two iPSC lines +/− SEM. (**D**) Six days post-infection, human RNA polymerase II (POLR2A) mRNA is increased 6.2-fold by HSV-1 and 22.0-fold by HSV-2. Bars represent values from two separate differentiations from one iPSC line +/− SEM. * *p* < 0.05, ** *p* < 0.01, *** *p* < 0.001 and **** *p* < 0.0001 from Student’s *t*-test.

## Data Availability

The data are available from the corresponding author upon request.

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
