# Peer review of "Herpes Simplex Virus 1 and 2 Infections during Differentiation of Human Cortical Neurons"

_viruses, 2021, doi:10.3390/v13102072_

Round 1

Reviewer 1 Report

HSV-1 and HSV-2 can cause serious disease of the central nervous system, but the effects of infection on differentiation and development of cortical neurons and the effects of neuronal differentiation on HSV virulence have not been well studied.

The authors used a novel in vitro model of human neuronal cell differentiation to study infection by HSV-1 and HSV-2 as the cells differentiated from stem cells to neural progenitor cells to cortical neurons.

My expertise is in lymphotropic alpha herpesviruses, rather than neurotropic alpha herpesviruses, so I am not familiar with neuronal cell culture, differentiation and activity and cannot comment in detail on this aspect of the work.

However, the study utilized a very comprehensive and thorough set of in vitro experiments, with appropriate controls, appropriate number of replicates, and thorough statistical analysis.  The manuscript is very well written, in a logical order, and is easy to read and understand – I enjoyed reading it and learned a lot.  The Introduction provides a logical and well referenced background to the study.  The Materials & Methods are sufficient in detail and the Results are clearly described with good well-presented figures and clear figure legends.  The authors have clearly explained the purpose of each part of the research.  The results are thoroughly discussed in the context of the current literature, unexpected results are discussed, and the conclusions are appropriate for the results.

These novel results will contribute to a better understanding of the mechanism of neuropathogenesis of HSV-1, HSV-2 and other neuropathogenic herpesviruses.

I have no major criticisms, just a list of suggestions for clarification and correction.

  • Line 148: What treatment was used for ‘mock infection’? It may be better to use the words ‘non-infected’.
  • Be consistent with use of ‘pfu’ or ‘PFU’.
  • Be consistent with use of the words ’medium’ and ‘media’.
  • Lines 264 and 489: Please explain the reason for treatment of cell cultures with ionizing radiation. Is it just to induce axonal damage as a positive control for increased production of NfL and tau proteins?  This was not clear to me.
  • Line 475: Change ‘that’ to ‘than’.
  • Figure 7 (C) and (D): How may replicates were there for each timepoint? Why are there no error bars for the pre-infection and 24-hour timepoints?

Reviewer 2 Report

In this study, the authors differentiated iPSCs into cortical neurons to study the effects of HSV-1 and -2 on cell viability, morphology, and gene expression while measuring the replication of viral progeny in differentiated neurons and precursors.  Similar to previous work, both viruses were found to negatively impact cellular function and viability, with HSV-2 typically more cytopathic.  This study employs a range of techniques, including electron microscopy to observe morphological changes and even viral trafficking, as well as qPCR and ELISA for a range of host gene products associated with neuronal dys/function. 

An interesting feature is the large effects on cell viability and synaptic activity observed rapidly after a low MOI (0.1) infection.  While LUHMES-derived neurons peak with a few hundred-fold increase in viral DNA levels (DOI: 10.1128/JVI.02210-18), the model employed in this study appears to support much greater levels of replication.  The ELISAs for tau are relevant to the burgeoning interest of HSV in amyloid-associated disorders.  While the rather large increase on POLR2A mRNA levels for HSV-2 is unexpected, 18S is a proper calibrator for this assay and several studies have shown that POLR2A is refractory to the transcriptional shutoff induced by HSV in epithelial cells.

The contrast to the previous study regarding Arc expression is addressed properly.  The discussion could benefit from addressing one additional paper (doi: 10.1371/journal.ppat.1008296) showing that HSV induces neuronal destruction, however direct comparisons of these two studies is again hindered by differences in species, cell origin, and viral strains.

The shift to neuronal models for HSV research is clinically important and this paper provides a solid framework for iPSC-derived cells in this regard while the field establishes which models best recapitulate in vivo phenotypes.  The choice of neurovirulent clinical isolates over more common lab-passaged strains is a strength. The techniques are sound and the conclusions drawn from them are supported by the data.  Other than mentioning the above reference from the Sawtell lab, I have only minor suggestions:

Lines 109, 213, 215: capitalize the L in “ml”

Line 121: ensure the indentation matches other paragraphs

Lines 182, 187: capitalize the L in “µl”

Please provide the qPCR information for POLR2A in section 2.7

Line 219: there appears to be an extra blank space after “Windehn”

Line 234: there appears to be an extra blank space after “of”

Lines 242, 243, 266: place a space between “°C” for consistency

The “A” label in Figure 3 is missing
